# Effect of quitting immediately vs progressively on smoking cessation for smokers at emergency department in Hong Kong: A posteriori analysis of a randomized controlled trial

**William Ho Cheung Li** [1]*, **Wei Xia**[2], **Man Ping Wang**[3], **Derek Yee Tak Cheung**[3], **Kai Yeung Cheung**[4], **Carlos King Ho Wong** [5], **Tai Hing Lam**[3]

1 Nethersole School of Nursing, The Chinese University of Hong Kong, Hong Kong, Hong Kong, 2 School of Nursing, Sun Yat-Sen University, Guangzhou, China, 3 School of Nursing, The University of Hong Kong, Hong Kong, Hong Kong, 4 United Christian Hospital, Hospital Authority, Hong Kong, Hong Kong, 5 Department of Pharmacology and Pharmacy & Department of Family Medicine and Primary Care, The University of Hong Kong, Hong Kong, Hong Kong

* williamli@cuhk.edu.hk

**Data Availability Statement:** All data belong to the funder (Health and Medical Research Fund, Food and Health Bureau, Hong Kong Special

## Abstract

### Background

A progressive approach to quitting smoking has been a popular strategy for motivating smokers who are reluctant to quit. However, whether this strategy can effectively achieve complete cessation or is as successful as quitting immediately remains unresolved. This study aimed to determine whether quitting immediately or progressively was more effective in achieving complete cessation among smokers in Hong Kong who presented at emergency departments.

### Methods and findings

A posteriori analysis of a single-blinded, multicenter, randomized controlled trial was performed. The original trials was conducted at emergency departments of four major acute hospitals in different districts of Hong Kong. In total, 1571 smokers 18 years or older who presented at 4 major emergency departments between July 4, 2015 and March 17, 2017 were randomized into an intervention group (n = 787) and a control group (n = 784). The intervention group received brief advice (about 1 minute) and could choose their own quit schedules (immediate or progressive, labeled QI and QP, respectively). The control group received a smoking cessation leaflet. Follow-ups were conducted at 1, 3, 6 and 12 months. The primary outcomes, by intention-to-treat, were biochemically validated abstinence between the QI subgroup and control group; between the QP subgroup and control group, and between the QI subgroup and QP subgroup at 6 months. After the propensity sore matching, the biochemically validated abstinence was statistically significantly higher in the QI subgroup than the control group at 6 months (12.1% vs 3.4%, *P* = 0.003; adjusted odds

Administrative Region). The data can be accessed only with the permission by the Bureau (https://rfs2.fhb.gov.hk/english/welcome/welcome.html). The authors did not have any special access privileges that others would not have.

**Funding:** This research was funded by Health and Medical Research Fund, Food and Health Bureau, Hong Kong Special Administrative Region (Dr Li: grant No. 12133111). The funder sources had no role in the design and conduct of the study; collection; management; analysis; and interpretation of the data; preparation; review; or approval of the manuscript; and decision to submit the manuscript for publication.

**Competing interests:** The authors have declared that no competing interests exist.

ratio [aOR] 4.34, 95% CI 1.63–11.52) and higher in the QP subgroup than the control group at 6 months (9.8% vs 3.4%, $P$ = 0.02; aORs 2.95, 95% CI: 1.04–8.39). No statistically significant differences of biochemically validated abstinence at both 6 month (12.1% vs 9.8%, $P$ = 0.49; aORs 1.50, 95% CI: 0.71–3.19) were found in the comparison between QI and QP subgroups.

## Conclusions

This study demonstrates that the strategy of quitting progressively is effective, especially for smokers who lack motivation or find it difficult to quit. If adopted routinely, such an approach can help achieve a greater level of smoking abstinence in the community.

## Trial registration

ClinicalTrials.gov: NCT02660957.

## Introduction

Cigarette smoking is addictive, and quitting the practice is very difficult [1, 2]. Our previous studies have found that many smokers recruited from outpatient clinics and the community were reluctant to quit, but showed an interest in reducing the number of cigarettes smoked per day [3–5]. Therefore, a potential strategy would be to allow or motivate smokers to quit progressively, with the ultimate goal of complete cessation of smoking. The progressive approach to quitting smoking has been used for a long time, following the assumption that smokers who reduce cigarette consumption and nicotine dependence will find it easier to further reduce the number of cigarettes smoked or quit smoking altogether [6, 7]. Nevertheless, whether the progressive approach can eventually lead to complete cessation or is as effective as abruptly quitting smoking remains controversial. Many previous studies have incorporated the strategy of quitting progressively in addition to nicotine replacement therapy, and their findings support the effectiveness of this strategy in achieving complete cessation in smokers who initially lacked the motivation to quit [3, 8]. However, not all smokers opt for pharmacotherapy to manage nicotine dependence. It has been reported that adherence to nicotine replacement therapy is low among Chinese smokers [9, 10]. It remains unclear whether using the progressive quitting strategy in such a population will help achieve long-term cessation [5, 7]. A previous trial conducted by our research group showed that quitting immediately was more effective than quitting progressively, although nicotine replacement therapy was not used. The outcomes of smokers were assessed at the 6-month medical follow-up in an outpatient clinic [4]. Another trial on smokers recruited from community settings showed that both the immediate and progressive approaches had similar 7-day point prevalence abstinence rates when assessed at the 6-month follow-up [5]. A Cochrane systematic review from 2019, which analyzed data from 22 randomized controlled trials (9219 participants) on quitting smoking immediately vs. progressively, found that neither approach was superior to the other in terms of long-term quitting rates [8]. Our previous trial examined the effectiveness of a brief self-determination theory-based smoking cessation intervention adopted for 1571 smokers who presented at emergency departments. We found that giving the smokers the option to either quit immediately or gradually doubled the quitting rates, compared to a control group that only received a smoking cessation pamphlet [11]. In this study, we aimed to conduct a

posteriori analysis of the data from this published randomized controlled trial to determine whether the smokers who chose to quit immediately or progressively had higher quitting rates than the smokers in the control group. In addition, the analysis aimed to determine which option (immediate or progressive) was more effective in achieving complete cessation.

## Materials and methods

### Study design and intervention

We analyzed the archived data from our previously published randomized controlled trial of a brief self-determination theory-based smoking cessation intervention adopted for smokers recruited from emergency departments [11]. Ethical approval was obtained from the Institutional Review Board of the University of Hong Kong/Hospital Authority Hong Kong West Cluster (UW14-528). The trial protocol has been published elsewhere [11]. Participants provided written informed consent.

Participants who presented at the emergency departments of four major acute care hospitals in different districts of Hong Kong were considered eligible if they were current smokers (occasional or daily) aged 18 years or older and triaged as either semi-urgent (level 4) or non-urgent (level 5) [12]. Exclusion criteria included an impaired mental status, cognitive impairment, communication barriers, or enrollment in other smoking cessation projects.

The sample size was calculated according to a previous trial [3] of a smoking reduction plus nicotine replacement therapy intervention involving 1154 Chinese adult smokers unwilling to quit smoking (biochemically validated quit rate of 4.4% [10 of 226] in the control group and 8.0% [74 of 928] in the intervention group at 6months). To detect a two-sided significant difference between groups by a chi-square test for comparing proportions with a power of 80% and significance level of 5%, the required sample size was estimated to be 1088 participants (544 in each group). Given an expected attrition rate of approximately 30% at the 6-month follow-up, the target was at least 1554 individuals (777 in each group). Between July 4, 2015 and March 17, 2017, 1571 smokers who presented at 4 major emergency departments consented to participate in this randomized controlled trial and were randomized into an intervention group (n = 787) and a control group (n = 784). Participants in the intervention group received brief advice and were given the option to either quit immediately (QI) or progressively (QP). Participants in the control group received a smoking cessation leaflet. Other details of the trial have been reported elsewhere [11]. Table 1 shows the characteristics of participants in the QI, QP, and the control groups. A Consolidated Standards of Reporting Trials (CONSORT) flowchart is presented in Fig 1.

### Measures

**Primary outcomes.** The primary outcome measures of this posterior analysis consisted of biochemically validated abstinence comparisons between the QI subgroup and control group, between the QP subgroup and control group, and between the QI and QP subgroups as assessed at the 6-month follow-up.

**Secondary outcomes.** The secondary outcomes included differences in biochemically validated abstinence as assessed at the 12-month follow-up, the self-reported 7-day point prevalence of abstinence as assessed at the 6- and 12-month follow-ups, and a self-reported reduction of at least 50% in daily cigarette consumption as assessed at the 6- and 12-month follow-ups between the QI subgroup and control group, between the QP subgroup and control group, and between the QI and QP subgroups.

**Table 1. Baseline characteristics of subjects in the Quit Immediately (QI) group, Quit Progressively (QP) group, and Control group in the original unmatched sample.**

| | QI (n = 242) | QP (n = 545) | Control (n = 784) | P value | Post-hoc analysis | Standardized differences | | |
| --- | --- | --- | --- | --- | --- | --- | --- | --- |
| | | | | | | QI vs QP | QI vs Control | QP vs Control |
| Age, mean(SD), y | 46.4(15.8) | 47.7(15.2) | 48.0(16.8) | 0.42 | | 0.084 | 0.098 | 0.019 |
| Gender | | | | 0.04 | QI>Con*# | 0.122 | 0.135 | 0.013 |
| Male | 202(83.5) | 484(88.8) | 700(89.3) | | | | | |
| Female | 40(16.5) | 61(11.2) | 84(10.7) | | | | | |
| Marital status | | | | 0.05 | QI>Con*## | 0.040 | 0.125 | 0.086 |
| Single/Divorce/Separate/Widowed | 104(43.0) | 221(40.6) | 278(35.5) | | | | | |
| Married/Cohabit | 138(57.0) | 324(59.4) | 506(64.5) | | | | | |
| Employment status | | | | 0.36 | | 0.017 | 0.048 | 0.065 |
| Unemployed/Retired | 63(26.0) | 137(25.1) | 224(28.6) | | | | | |
| Employed | 179(74.0) | 408(74.9) | 560(71.4) | | | | | |
| Educational level | | | | 0.11 | | 0.115 | 0.048 | 0.074 |
| Tertiary | 21(8.7) | 27(5.0) | 56(7.1) | | | | | |
| Secondary or below | 221(91.3) | 518(95.0) | 728(92.9) | | | | | |
| Monthly household income, US $ | | | | 0.09 | | 0.115 | 0.142 | 0.026 |
| ≥3825 (HKD 30000) | 22(9.1) | 73(13.4) | 114(14.5) | | | | | |
| <3825 (HKD 29999) | 220(90.9) | 472(86.6) | 670(85.5) | | | | | |
| Smoking-related chronic disease | | | | 0.18 | | 0.050 | 0.110 | 0.058 |
| Yes | 12(5.0) | 35(6.4) | 64(8.2) | | | | | |
| No | 230(95) | 510(93.6) | 720(91.8) | | | | | |
| Health utility score by SF-6D[a] | 0.59(0.1) | 0.57(0.1) | 0.57(0.1) | 0.08 | | 0.200 | 0.200 | 0.000 |
| Daily cigarette consumption | 12.8(7.5) | 15.0(7.9) | 13.5(7.6) | 0.001 | QI>QP*# | 0.286 | 0.093 | 0.195 |
| Nicotine dependence by Heaviness of Smoking Index (HIS)[b] | | | | <0.001 | QI< QP*# | 0.271 | 0.168 | 0.102 |
| Moderate to heavy(3–6) | 95(39.3) | 303(55.6) | 387(49.4) | | QI< Con*# | | | |
| Light(≤2) | 147(60.7) | 242(44.4) | 397(50.6) | | | | | |
| Age at starting smoking weekly | 17.4(6.2) | 17.2(5.7) | 17.6(6.5) | 0.52 | | 0.034 | 0.031 | 0.067 |
| Tried to quit smoking for more than 24 hours | | | | <0.001 | QI>QP*# | 0.262 | 0.257 | 0.005 |
| Yes | 192(79.3) | 357(65.5) | 516(65.8) | | QI>Con*# | | | |
| No | 50(20.7) | 188(34.5) | 268(34.2) | | | | | |
| Tried to reduce smoking for more than 24 hours | | | | 0.77 | | 0.000 | 0.029 | 0.029 |
| Yes | 119(49.2) | 268(49.2) | 400(51.0) | | | | | |
| No | 123(50.8) | 277(50.8) | 384(49.0) | | | | | |
| Readiness to quit | | | | 0.34 | | 0.084 | 0.022 | 0.063 |
| Quit ≤ 30 days | 71(29.3) | 133(24.7) | 218(28.1) | | | | | |
| Quit > 30 days | 171(70.7) | 405(75.3) | 557(71.9) | | | | | |
| Self-efficacy against tobacco by SEQ-12[c] | 29.8(12.3) | 29.1(10.8) | 28.2(11.3) | 0.14 | | 0.060 | 0.135 | 0.078 |

Continuous variables are reported as mean ± standard deviation. Dichotomous variables are reported as N (Percent).

Abbreviations: SF-6D, Shot-Form Six-Dimension; SEQ-12, Smoking Self-Efficacy Questionnaire.

[a]The SF-6D is composed of 6 multilevel dimensions. The SF-6D scores were weighted from a sample of the general population, which ranged from 0 to 1.

[b]The Heaviness of Smoking Index, a 2-item index from multiple-choice response options (0–3), was determined by assessing cigarettes smoked per day and time to smoke after waking; the higher the indexes, the greater smoking nicotine dependence.

[c]On a 12-item 5-point Likert-type scale in the SEQ-12, responses ranged from "not at all sure" to "absolutely sure." A summary score of the SEQ-12 ranged from 12 to 60, with higher scores indicating higher self-efficacy.

*: $P< 0.05$.

#: Using the Tukey's honestly significant difference post-hoc test as equal Variances assumed.

##: Using the Games -Howell post-hoc test as equal Variances not assumed.

## Statistical analysis

Data analysis was performed using the IBM Statistical Package for Social Sciences (SPSS) for Windows (version 25.0; IBM). To minimize the effects of potential confounding factors (demographic characteristics and smoking history of participants) on the primary and secondary outcome measures, a three-way propensity matched analysis was performed. To estimate the propensity scores, all demographic and smoking variables were included in the multinomial regression to maximally inform the propensity of the dependent variables [13]. The QI subgroup vs. control group and QP subgroup vs. control group were matched 1:1 using a nearest-neighbor approach with caliper restrictions [14]. A three-way matched data set was then created without replacement by extracting participants from the QI or QP subgroup who had common matches with participants in the control group [14, 15]. The standardized differences in demographic and smoking variables were compared to diagnose the balancing of the matched groups [16]. For continuous and dichotomous variables, the standardized difference used is shown in the S1 and S2 Figs, respectively [17].

The baseline characteristics of the participants in the QI, QP, and control groups were compared using an analysis of variance (ANOVA) for continuous variables, the chi-square test for categorical variables, and the two-tailed Fisher's exact test based on the group cell size. For the variables showing significant difference in ANOVA, the Tukey's honestly significant difference post-hoc test the Games -Howell post-hoc test and were performed when the assumption of equal variances was met and not met, respectively. All analyses were performed based on intention-to-treat, in which participants lost to follow-up were assumed to be active smokers with no changes with respect to the baseline. For primary analysis, the differences in biochemically validated quit rates, as assessed at the 6-month follow-up, between the QI, QP, and control groups were analyzed using the propensity score matched samples. A similar approach was used to analyze the differences in secondary outcomes.

Univariate logistic regression was performed to examine the crude odds ratios (ORs) for primary and secondary outcomes using both the original unmatched and matched samples. A Generalized Logistic Mixed Model (GLMM) was then used to calculate the adjusted odds ratios (aORs) for primary and secondary outcomes after adjusting for characteristics at baseline and the random effect of hospitals using the matched sample. A P value < 0.05 was considered to be statistically significant.

In addition, a posteriori analysis was performed to examine the association between the quantity of smoking reduction across all follow-ups and abstinence at the final follow-up. The percentage reduction was calculated by dividing the difference in daily cigarette consumption between the baseline and a given follow-up by the number of cigarettes consumed at baseline. Multiple logistic regression models were used to examine the predictive power of the absolute and percentage reductions on 12-month abstinence in participants who had not quit by the time of the follow-ups. Each model examined the reduction quantity at a given follow-up as either the absolute or percentage reduction to predict the 12-month abstinence. All models were adjusted for the treatment condition (QI, QP, and control group), demographic and smoking characteristics at the baseline, and the random effect of hospitals. The observed power $(1-\beta)$ of quitting immediately and quitting progressively on the biochemically validated quit rate, the self-reported quit rate, and self-reported reduction of cigarette consumption were then calculated using G*power. A scatterplot and fitted line analysis were then used to demonstrate the linear association between the absolute or percentage cigarette reduction at 1-, 3-, and 6-month follow-ups and biochemically validated abstinence as assessed at the 12-month follow-up. Given the discrepancies in smoking profiles between the QI and QP subgroups, a two-group propensity matching between the QP and control group was conducted

to provide more information on the outcomes in smokers who chose to quit smoking progressively. Similar analyses as described above were also additionally performed.

## Results

Fig 1 shows that in the intervention group, 242 participants (30.7%) chose to quit smoking immediately and 545 participants (69.3%) chose to quit smoking progressively. Compared

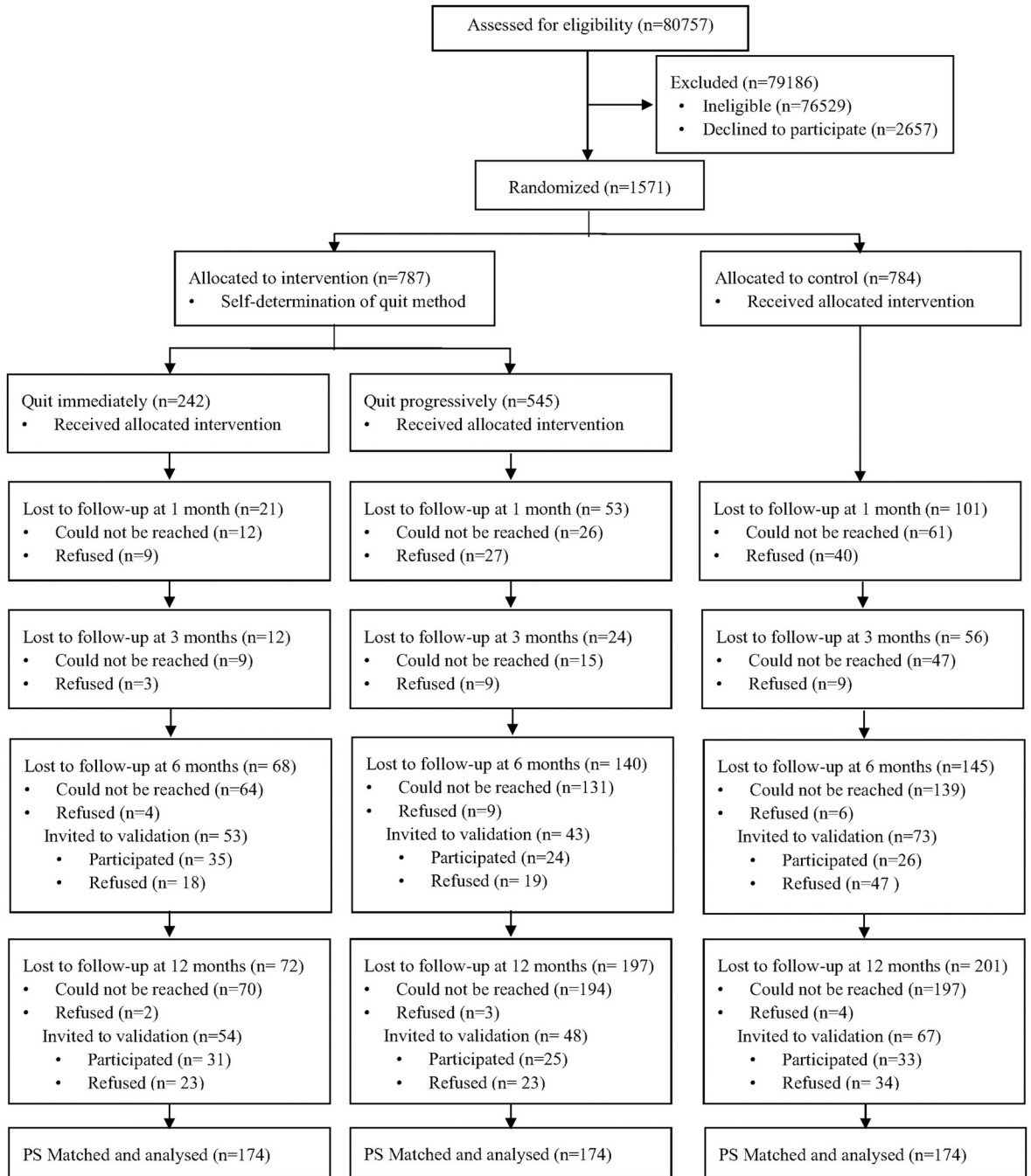

**Fig 1. CONSORT flowchart.**

with the QI subgroup, the QP subgroup had a significantly higher mean rate of daily cigarette consumption (QP vs. QI: 15.0 vs. 12.8), more moderate to heavy nicotine dependence (QP vs. QI: 55.6% vs. 39.3%), and a higher number of participants who had not previously attempted to quit smoking (QP vs. QI: 34.5% vs. 20.7%). After propensity score matching, 174 pairs of subjects in the QI, QP, and control groups were matched and analyzed. In the matched sample shown in Table 2, the absolute standardized differences in all covariates were less than 0.10, and the means and prevalence of baseline covariates were similar in the two matched samples, indicating good balance between the groups [18].

Tables 3 and 4 showed that after propensity score matching, the biochemically validated abstinence was significantly higher in the QI subgroup than in the control group as assessed at the 6-month (12.1% vs. 3.4%, $P$ = 0.003; aOR = 4.34, 95% CI: 1.63–11.52) and 12-month follow-ups (10.9% vs. 4.0%, $P$ = 0.01; aOR = 3.23, 95% CI: 1.24–8.43). The number needed to treat (NNT) for the QI subgroup was 11.5 [1/(0.121–0.034)]. Compared with the control group, the QI subgroup showed a significantly higher self-reported 7-day point prevalence of abstinence as assessed at the 6-month (21.8% vs. 7.5%, $P$ < 0.001; aOR: 4.34, 95% CI: 1.63–11.52) and 12-month follow-ups (20.7% vs 6.3%, $P$ < 0.001; aOR: 3.23, 95% CI: 1.24–8.43). After excluding those participants who completely ceased smoking, the number of participants who self-reported a reduction in smoking of at least 50% was found to be significantly higher in the QI subgroup than control group at both the 6-month (19.9% vs. 10.6%, $P$ = 0.03; aOR = 2.15, 95% CI: 1.10–4.24), and 12-month follow-ups (18.8% vs. 10.4%, $P$ = 0.04; aOR = 1.95, 95% CI: 0.96–3.93). However, after adjusting for demographics and smoking characteristics at the baseline and the random effect of hospitals, the aOR as assessed at the 12-month follow-up was no longer significantly different between the two groups ($P$ = 0.07; aOR = 1.95, 95% CI: 0.96–3.93).

The biochemically validated abstinence was also significantly higher in the QP subgroup than the control group when it was measured at the 6-month (9.8% vs. 3.4%, $P$ = 0.02; aOR = 2.95, 95% CI: 1.04–8.39) and 12-month follow-ups (10.3% vs. 4.0%, $P$ = 0.02; aOR = 2.85, 95% CI: 1.11–7.33). The NNT for the QP subgroup was 15.6 [1/(0.098–0.034)]. Compared with the control group, the QP subgroup showed a significantly higher self-reported 7-day point prevalence of abstinence when measured at the 6-month (14.4% vs. 7.5%, $P$ = 0.04; aOR = 1.96, 95% CI: 1.12–4.08) and 12-month follow-ups (19.0% vs. 6.3%, $P$ < 0.001; aOR = 3.10, 95% CI: 1.52–6.79). After excluding those participants who completely ceased smoking, the number of participants who self-reported a reduction in smoking of at least 50% was significantly higher in the QP subgroup than in the control group as measured at the 6-month (24.2% vs. 10.6%, $P$ = 0.001; aOR = 2.70, 95% CI: 1.40–5.23) and 12-month follow-ups (29.8% vs. 10.4%, $P$ < 0.001; aOR = 3.42, 95% CI: 1.76–6.64). A comparison of the baseline characteristics and smoking profiles between the QP and control groups after two-group propensity score matching is presented in the S1 Table. The cessation outcomes showed that the biochemically validated and self-reported abstinence rates among subjects in the matched QP group were significantly higher than those in the matched control group as assessed at both the 6- and 12-month follow-ups (S2 Table). After excluding those participants who completely ceased smoking, participants who self-reported a reduction in smoking of at least 50% was higher in the QP subgroup than in the control group. This increase was significantly different when measured at the 12-month follow-up, but not at the 6-month follow-up.

There were no significant differences in biochemically validated abstinence between the QI and QP subgroups at when assessed at the either 6-month (12.1% vs. 9.8%, $P$ = 0.49; aOR = 1.50, 95% CI: 0.71–3.19) or 12-month follow-up (10.9% vs. 10.3%, $P$ = 0.86; aOR = 1.22, 95% CI: 0.57–2.59). Higher self-reported abstinence was reported in the QI subgroup than in the QP subgroup, but this difference was not significant as assessed at either the 6-month

**Table 2. Baseline characteristics of subjects in the Quit Immediately (QI) group, Quit Progressively (QP) group, and Control group in the propensity-score matched sample.**

| | QI (n = 174) | QP (n = 174) | Con (n = 174) | P value | Standardized differences | | |
| --- | --- | --- | --- | --- | --- | --- | --- |
| | | | | | QI vs QP | QI vs Con | QP vs Con |
| Age, mean(SD), y | 46.6(15.6) | 47.0(16.2) | 47.1(15.6) | 0.96 | 0.025 | 0.045 | 0.019 |
| Gender | | | | 0.71 | 0.067 | 0.067 | 0.000 |
| Male | 147(84.5) | 152(87.4) | 151(86.8) | | | | |
| Female | 27(15.5) | 22(12.6) | 23(13.2) | | | | |
| Marital status | | | | 0.95 | 0.010 | 0.018 | 0.008 |
| Single/Divorce/Separate/Widowed | 72(41.4) | 73(42.0) | 75(43.1) | | | | |
| Married/Cohabit | 102(58.6) | 101(58.0) | 100(56.9) | | | | |
| Employment status | | | | 0.26 | 0.012 | 0.076 | 0.064 |
| Unemployed/Retired | 40(23) | 41(23.6) | 52(29.9) | | | | |
| Employed | 134(77) | 133(76.4) | 122(70.1) | | | | |
| Educational level | | | | 0.13 | 0.000 | 0.089 | 0.089 |
| Tertiary | 14(8.0) | 14(8.0) | 9(5.2) | | | | |
| Secondary or below | 160(92.0) | 160(92.0) | 165(97.1) | | | | |
| Monthly household income, US $ | | | | 0.79 | 0.061 | 0.014 | 0.046 |
| ≥3825 (HKD 30000) | 17(9.8) | 21(12.1) | 19(10.9) | | | | |
| <3825 (HKD 29999) | 157(90.2) | 153(87.9) | 155(89.2) | | | | |
| Smoking-related chronic disease | | | | 0.96 | 0.000 | 0.022 | 0.022 |
| Yes | 9(5.2) | 9(5.2) | 8(4.6) | | | | |
| No | 165(94.8) | 165(94.8) | 166(95.4) | | | | |
| Health utility score by SF-6D[a] | 0.58(0.1) | 0.58(0.1) | 0.59(0.1) | 0.47 | 0.000 | 0.090 | 0.090 |
| Daily cigarette consumption | 13.8(7.8) | 13.9(7.4) | 13.9(7.6) | 0.99 | 0.013 | 0.013 | 0.000 |
| Nicotine dependence by Heaviness of Smoking Index (HIS)[b] | | | | 0.55 | 0.067 | 0.095 | 0.028 |
| Moderate to heavy(3–6) | 78(44.8) | 85(48.9) | 88(50.6) | | | | |
| Light(≤2) | 96(55.2) | 89(51.1) | 86(49.4) | | | | |
| Age at starting smoking weekly | 17.6(6.4) | 17.2(5.5) | 17.6(6.2) | 0.84 | 0.067 | 0.000 | 0.067 |
| Tried to quit smoking for more than 24 hours | | | | 0.73 | 0.066 | 0.000 | 0.066 |
| Yes | 132(75.9) | 126(72.4) | 132(75.9) | | | | |
| No | 42(24.1) | 48(27.6) | 42(24.1) | | | | |
| Tried to reduce smoking for more than 24 hours | | | | 0.95 | 0.010 | 0.028 | 0.018 |
| Yes | 80(46.0) | 81(46.6) | 83(47.7) | | | | |
| No | 94(54.0) | 93(53.4) | 91(52.3) | | | | |
| Readiness to quit | | | | 0.93 | 0.022 | 0.009 | 0.032 |
| Quit ≤ 30 days | 45(25.9) | 43(24.7) | 46(26.4) | | | | |
| Quit > 30 days | 129(74.1) | 131(75.3) | 128(73.6) | | | | |
| Self-efficacy against tobacco by SEQ-12[c] | 29.3(12.0) | 29.0(10.4) | 28.4(10.5) | 0.73 | 0.027 | 0.078 | 0.053 |

Continuous variables are reported as mean ± standard deviation. Dichotomous variables are reported as N (Percent).

Abbreviations: SF-6D, Shot-Form Six-Dimension; SEQ-12, Smoking Self-Efficacy Questionnaire.

[a]The SF-6D is composed of 6 multilevel dimensions. The SF-6D scores were weighted from a sample of the general population, which ranged from 0 to 1.

[b]The Heaviness of Smoking Index, a 2-item index from multiple-choice response options (0–3), was determined by assessing cigarettes smoked per day and time to smoke after waking; the higher the indexes, the greater smoking nicotine dependence.

[c]On a 12-item 5-point Likert-type scale in the SEQ-12, responses ranged from "not at all sure" to "absolutely sure." A summary score of the SEQ-12 ranged from 12 to 60, with higher scores indicating higher self-efficacy.

**Table 3. Cessation outcomes of subjects in the Quit Immediately (QI) group, Quit Progressively group and control group in the original unmatched sample and the propensity-score matched sample.**

| | Original unmatched sample | | | | | Propensity-score matched sample | | | P value | Post-hoc analysis |
|---|---|---|---|---|---|---|---|---|---|---|
| | N (%) | | | | | N (%) | | | | |
| | QI (n = 242) | QP (n = 545) | Con (n = 787) | P value | Post-hoc analysis | QI (n = 174) | QP (n = 174) | Con (n = 174) | | |
| Biochemically validated abstinence | | | | | | | | | | |
| 6 months | 34 (14.0) | 19 (3.5) | 22 (2.8) | <0.001 | QI>QP***# | 21 (12.1) | 17 (9.8) | 6 (3.4) | 0.01 | QI>Com**# |
| | | | | | QI>Con***# | | | | | QP>Con*# |
| 12 months | 31 (12.8) | 24 (4.4) | 33 (4.2) | <0.001 | QI>QP***# | 19 (10.9) | 18 (10.3) | 7 (4.0) | 0.04 | QI>Com**# |
| | | | | | QI>Con***# | | | | | QP>Con*# |
| Self-reported 7-day point prevalence of abstinence | | | | | | | | | | |
| 6 months | 53 (21.9) | 43 (7.9) | 73 (9.3) | <0.001 | QI>QP***# | 38 (21.8) | 25 (14.4) | 13 (7.5) | 0.001 | QI>Com**# |
| | | | | | QI>Con***# | | | | | QP>Con*# |
| 12 months | 54 (22.3) | 48 (8.8) | 67 (8.5) | <0.001 | QI>QP***# | 36 (20.7) | 33 (19.0) | 11(6.3) | 0.001 | QI>Com**# |
| | | | | | QI>Con***# | | | | | QP>Com**# |
| Self-reported reduction of ≥ 50% in cigarette consumption[a] | | | | | | | | | | |
| 6 months | 34 (18.0) | 89 (17.7) | 127 (17.9) | 0.99 | | 27 (19.9) | 36 (24.2) | 17(10.6) | 0.006 | QI>Com**# |
| | | | | | | | | | | QP>Con***# |
| 12 months | 34 (18.1) | 96 (19.3) | 105 (14.6) | 0.09 | | 26 (18.8) | 42 (29.8) | 17(10.4) | <0.001 | QI>QP*# |
| | | | | | | | | | | QI>Com*# |
| | | | | | | | | | | QP>Con***# |

Subjects lost to follow-up were assumed to be active smokers with no changes from baseline.

[a] The quitters were excluded in both numerators and denominators.

\*: P<0.05

\*\*: P<0.01

\*\*\*: P<0.001

#: Using the Tukey's honestly significant difference post-hoc test as equal Variances assumed.

##: Using the Games -Howell post-hoc test as equal Variances not assumed.

(21.8% vs. 14.4%, $P = 0.07$; aOR = 1.67, 95% CI: 0.93–2.99) or 12-month follow-up (20.7% vs. 19.0%, $P = 0.69$; aOR = 1.08, 95% CI: 0.62–1.87). Excluding those participants who completely ceased smoking, the number of participants who self-reported a reduction in smoking of at least 50% was lower in the QI subgroup than in the QP subgroup. This reduction was significantly different between the two groups when assessed at the 12-month follow-up (18.8% vs. 29.8%, $P = 0.03$; aOR = 0.60, 95% CI: 0.31–3.98), but not at the 6-month follow-up (19.9% vs. 24.2%, $P = 0.38$; aOR = 0.77, 95% CI: 0.42–1.39). Table 5 presented that the powers of the quitting immediately had a and quitting progressively on the biochemically validated quit rate, the self-reported quit rate, and self-reported reduction of cigarette consumption were acceptable (all larger than 0.80) to detect the hypothesis in this study.

The scatterplot and fitted line analysis showed that the values of both the absolute and percent cigarette reduction at the 1-, 3-, and 6-month follow-ups were associated with the biochemically validated abstinence as assessed at the 12-month follow-up (Fig 2). The $R^2$ showed that the percent cigarette reduction (a-2, b-2, c-2) could better predict the 12-month abstinence than the absolute cigarette reduction (a-1, b-1, c-1).

## Discussion

The results of this a posteriori analysis showed that the number of smokers in the intervention group who chose to quit smoking progressively outnumbered that of smokers who chose to

**Table 4. Logistic regression for validated abstinence, self-report abstinence, and reduction of cigarette consumption among the QI group (n = 174), QP group (n = 174) and control group (n = 174) in the propensity-score matched sample.**

| | QI group vs QP group | | | | QI group vs Control group | | | | QP group vs Control group | | | |
|---|---|---|---|---|---|---|---|---|---|---|---|---|
| | Crude OR[a] | P value | Adjusted OR[b] | P value | Crude OR[a] | P value | Adjusted OR[b] | P value | Crude OR[a] | P value | Adjusted OR[b] | P value |
| Biochemically validated abstinence | | | | | | | | | | | | |
| 6 months | 1.40(0.70, 2.83) | 0.29 | 1.50(0.71, 3.19) | 0.29 | 2.88(1.13, 7.32) | 0.005 | 4.34(1.63, 11.52) | 0.003 | 3.03(1.17, 7.89) | 0.02 | 2.95(1.04, 8.39) | 0.02 |
| 12 months | 1.19(0.59, 2.39) | 0.59 | 1.22(0.57, 2.59) | 0.59 | 2.34(1.16, 5.72) | 0.019 | 3.23(1.24, 8.43) | 0.01 | 2.75(1.12, 6.77) | 0.03 | 2.85(1.11, 7.33) | 0.03 |
| Self-reported 7-day point prevalence of abstinence | | | | | | | | | | | | |
| 6 months | 1.52(0.88, 2.65) | 0.07 | 1.67(0.93, 2.99) | 0.07 | 2.29(1.17, 4.48) | <0.001 | 3.65(1.80, 7.43) | <0.001 | 2.08(1.03, 4.21) | 0.04 | 1.96(1.12, 4.08) | 0.04 |
| 12 months | 1.10(0.65, 1.85) | 0.69 | 1.08(0.62, 1.87) | 0.69 | 2.17(1.09, 4.34) | <0.001 | 3.85(1.82, 8.16) | <0.001 | 3.47(1.69, 7.12) | 0.001 | 3.10(1.52, 6.79) | 0.002 |
| Self-reported reduction of ≥ 50% in cigarette consumption | | | | | | | | | | | | |
| 6 months | 0.78(0.44, 1.37) | 0.38 | 0.77(0.42, 1.39) | 0.38 | 2.10(1.09, 4.04) | 0.027 | 2.15(1.10, 4.24) | 0.03 | 2.70(1.44, 5.05) | <0.002 | 2.70(1.40, 5.23) | 0.001 |
| 12 months | 0.55(0.31, 0.96) | 0.03 | 0.60(0.31, 0.98) | 0.03 | 1.99 (1.03, 3.85) | 0.040 | 1.95(0.96, 3.93) | 0.07 | 3.64(1.96, 6.76) | <0.001 | 3.42(1.76, 6.64) | <0.001 |

Subjects lost to follow-up were assumed to be active smokers with no changes from baseline.

[a] Crude RR = Crude Relative Risks. Crude estimates from the univariable logistic regression

[b] Adjusted RR = Adjusted Relative Risks. Adjusted estimates from Generalized Logistic Mixed Model adjusted for age, sex, employment status, health utility score, daily cigarette consumption, nicotine dependence level, readiness to quit, and smoking efficacy against tobacco at baseline, and random effect of hospitals.

quit smoking immediately by more than two folds (progressive vs. immediate: 69.3% vs. 30.7%). The results also indicated that smokers in the QP subgroup had significantly higher rates of daily cigarette consumption and nicotine dependency and had made fewer attempts to quit previously than those in the QI subgroup. The findings of this study provide support to existing reports in the literature which show that many heavy smokers are reluctant to quit smoking immediately. Therefore, smoking and the quitting histories of smokers should be considered when recommending different types of smoking cessation interventions. For heavy or hard-core smokers who are reluctant to quit, intervention strategies that enforce immediate quitting may be perceived as being too harsh and be ineffective in helping them to cease smoking. In contrast, those who smoke a few cigarettes a day with mild nicotine dependence may deem the progressive quitting approach unnecessary or superfluous and consequently undermine the effectiveness of the approach.

**Table 5. Posterior power calculation for quitting immediately (QI) and quitting progressively (QP) compared to smoking cessation leaflet (Control group).**

| | Observed power (1-β) | |
|---|---|---|
| | Quitting Immediately | Quitting Progressively |
| Biochemically validated abstinence | | |
| 6 months | 0.91 | 0.84 |
| 12 months | 0.85 | 0.82 |
| Self-reported 7-day point prevalence of abstinence | | |
| 6 months | 0.96 | 0.81 |
| 12 months | 0.97 | 0.95 |
| Self-reported reduction of ≥ 50% in cigarette consumption | | |
| 6 months | 0.82 | 0.92 |
| 12 months | 0.80 | 0.98 |

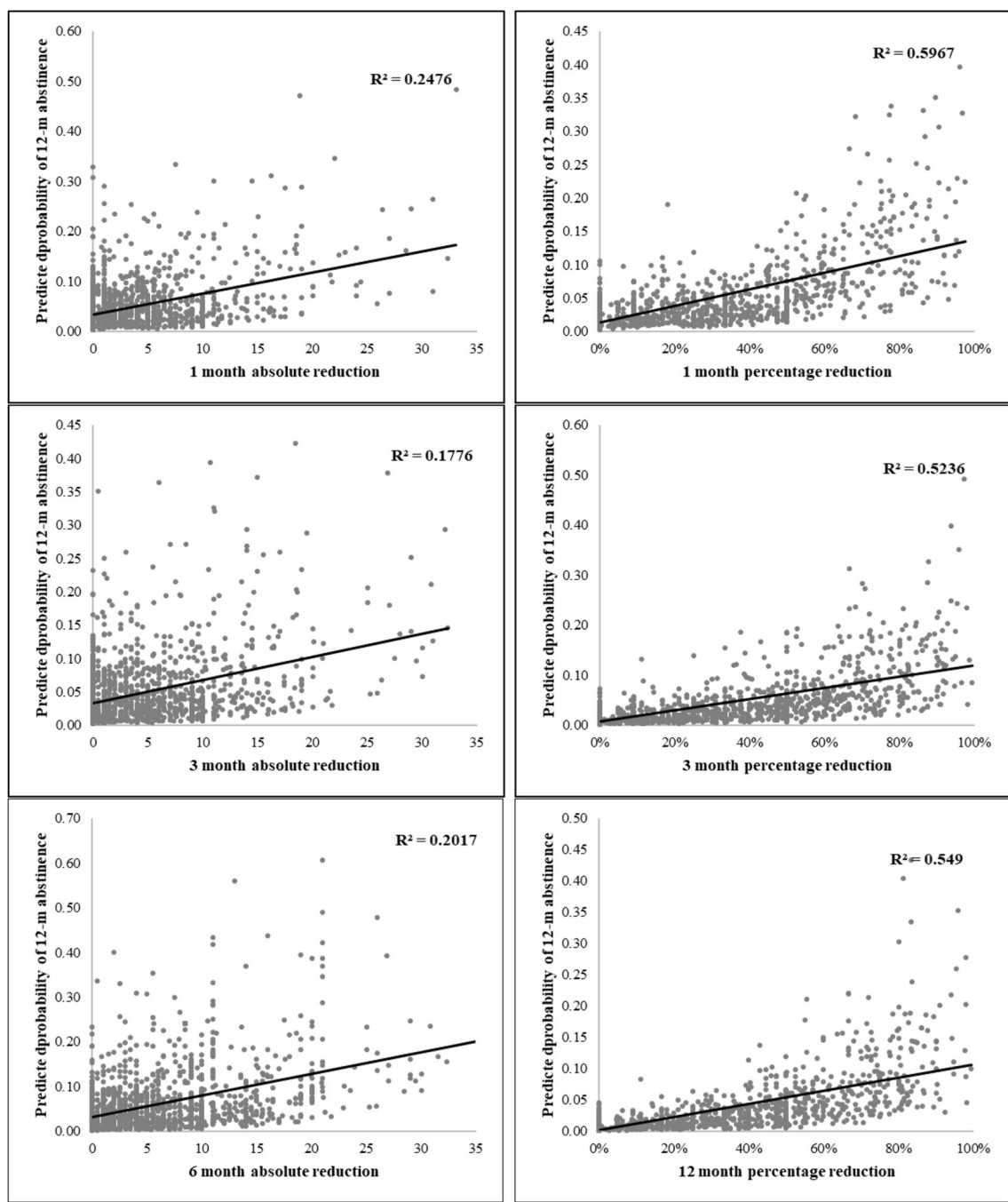

**Fig 2. Scatterplot of the predicted probability of 12-month abstinence against the reduction of cigarette consumption at 1-month, 3-month, and 6-month.**

The subgroup analyses showed that smokers in both the QI and QP subgroups had significantly higher biochemically validated abstinence and self-reported 7-day point prevalence of abstinence rates at the 6- and 12-month follow-ups than those in the control group. The results demonstrated that offering a brief smoking cessation intervention to smokers and allowing them to choose the quitting schedules effectively promoted the cessation of smoking. In addition, a significantly higher proportion of smokers who had not quit in the QP subgroup achieved at least

50% reduction in cigarette consumption by the 12-month follow-up, compared to those in the QI subgroup. Though the ultimate goal of quitting progressively is the complete cessation of smoking, it is anticipated that these smokers will find it much easier to gradually reduce their cigarette consumption or quit smoking altogether in the near future, given that they have already initiated the process and reduced their nicotine dependence [6, 7]. The fitted line analysis supported the potential of using the rates of reduction in cigarette consumption to predict future abstinence from smoking. These findings imply that progressive quitting is a useful alternative approach for smokers who lack motivation and experience difficulty in quitting smoking [19].

## Limitations

Given the discrepancies in the smoking profiles of smokers who choose to quit either immediately or progressively, it is difficult to compare the smoking cessation outcomes between the QI and QP groups. Therefore, three-group and two-group propensity score matching analyses were conducted. This study conducted a posteriori analysis of data from a previous trial, which could not provide sufficient evidence for a causal relationship between a reduction in smoking and abstinence from smoking. In future, a randomized controlled trial should be conducted in which smokers with similar smoking profiles should be recruited to test the differences in reduction and abstinence for longer follow-up periods.

## Implications for clinical practice

This study addresses the important question of whether the approach to quitting smoking in a progressive manner is or is not effective. Our findings also addressed existing gaps in the field by demonstrating that quitting progressively is effective, especially for chronic smokers who lack motivation or find it difficult to quit. A measured application of these results can help achieve a greater level of abstinence from smoking and make important contributions to evidence-based practice. Most importantly, the outcomes of this original study can inform future researchers and policymakers on designing effective smoking cessation interventions and policies for smokers who are reluctant to quit smoking immediately. Thus, the study has important implications for clinical practice and the improvement of public health. In future, we will explore the means to retain smokers in gradual cessation programs as they reduce their frequency of smoking, develop more successful methods to encourage reduction in smoking, and find ways to prevent a perception of failure by participants, which usually causes them abandon their attempts to reduce the number of cigarettes smoked and quit smoking. Finally, in Hong Kong, which is a region with a low prevalence of smoking yet has many hard-core smokers, our results can guide future strategies toward a total ban on tobacco sales.

## Conclusions

This secondary analysis of a randomized controlled trial provides further support to a previous study that allowed smokers to choose their quitting schedules, which was essential in motivating them to quit smoking. This study supplements the previous trial by determining that the progressive quitting approach is effective, especially for smokers who lack motivation or find it difficult to quit.

## Supporting information

**S1 Checklist. CONSORT 2010 checklist of information to include when reporting a randomized trial[a].**
(PDF)

**S1 Fig. Calculation formula of the standardized difference for continuous variables.**
(TIF)

**S2 Fig. Calculation formula of the standardized difference for dichotomous variables.**
(TIF)

**S1 Table. Comparison of baseline characteristics and smoking profiles among subjects in the QP group and control group in the original unmatched sample and the propensity-score matched sample.**
(DOCX)

**S2 Table. Cessation outcomes of subjects in the QP group vs. control group original trial protocol and statistical analysis plan.**
(DOCX)

**S1 File. Trial protocol with statistical analysis plan.**
(PDF)

## Author Contributions

**Conceptualization:** William Ho Cheung Li, Wei Xia, Man Ping Wang, Derek Yee Tak Cheung, Tai Hing Lam.

**Data curation:** Wei Xia, Carlos King Ho Wong.

**Formal analysis:** William Ho Cheung Li, Wei Xia, Carlos King Ho Wong.

**Funding acquisition:** William Ho Cheung Li.

**Methodology:** William Ho Cheung Li, Wei Xia, Man Ping Wang, Derek Yee Tak Cheung, Tai Hing Lam.

**Project administration:** William Ho Cheung Li, Kai Yeung Cheung.

**Resources:** William Ho Cheung Li, Kai Yeung Cheung.

**Supervision:** William Ho Cheung Li, Wei Xia, Tai Hing Lam.

**Writing – original draft:** William Ho Cheung Li, Wei Xia.

**Writing – review & editing:** William Ho Cheung Li, Wei Xia, Man Ping Wang, Derek Yee Tak Cheung, Kai Yeung Cheung, Carlos King Ho Wong, Tai Hing Lam.

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
