## [Decision Letter · Decision Letter 0]

26 Apr 2022

PONE-D-21-03778

Effect of Quitting Immediately vs Progressively on Smoking Cessation for Smokers at Emergency Department in Hong Kong: A Posteriori Analysis of a Randomized Controlled Trial

PLOS ONE

Dear Dr. Li,

Thank you for submitting your manuscript to PLOS ONE. After careful consideration, we feel that it has merit but does not fully meet PLOS ONE’s publication criteria as it currently stands. Therefore, we invite you to submit a revised version of the manuscript that addresses the points raised during the review process.

We look forward to receiving your revised manuscript.

Kind regards,

Hao Xue

Academic Editor

PLOS ONE

Journal Requirements:

Reviewers' comments:

Reviewer's Responses to Questions

**Comments to the Author**

1. Is the manuscript technically sound, and do the data support the conclusions?

Reviewer #1: Partly

2. Has the statistical analysis been performed appropriately and rigorously? 

Reviewer #1: Yes

3. Have the authors made all data underlying the findings in their manuscript fully available?

Reviewer #1: No

4. Is the manuscript presented in an intelligible fashion and written in standard English?

Reviewer #1: Yes

5. Review Comments to the Author

Reviewer #1: A two-arm randomized controlled study aimed to determine which method of quitting smoking is more effective: suddenly or progressively. In a propensity score matched analysis, the 6-month abstinence rate was statistically significantly higher in the intervention group who chose to quit smoking immediately compared to controls. At 6-months, the cessation of smoking rate was also higher in the intervention group who chose to quit smoking progressively compared to controls. The smoking cessation rates were not statistically significantly different in the two intervention groups.

Minor revisions:

1- Tables 1 to 3: For each baseline characteristic, provide only one p-value for comparing all three groups. If the p-value is significant, perform step-down tests to determine pairwise differences.

2-Page 9: Explain the rationale for using a GEE model when adjusting for baseline characteristics rather than applying multivariate logistic regression models.

3- Page 10, Line 1: The figure number had been omitted.

4- Provide a post-hoc power calculation.

5- Explain the graphical results in figure 2.

6. PLOS authors have the option to publish the peer review history of their article (what does this mean?). If published, this will include your full peer review and any attached files.

Reviewer #1: No

---

## [Author Response · Author response to Decision Letter 0]

19 May 2022

Dear Editor,

Re: "Effect of Quitting Immediately vs Progressively on Smoking Cessation for Smokers at Emergency Department in Hong Kong: A Posteriori Analysis of a Randomized Controlled Trial" (PONE-D-21-03778)

We sincerely thank the editor and reviewer’s very useful and constructive comments. We have made changes accordingly. Please also refer to the following point-by-point responses to the comments from the reviewer and editor. 

Please kindly let me know in case any responses are not clear or the information is not adequate to clarify concerns. Thanks for giving us an opportunity to revise the manuscript. 

Please also note that all data are belong to the funder (Health and Medical Research Fund, Food and Health Bureau, Hong Kong Special Administrative Region). The data can be accessed only with the permission by the Bureau (https://rfs2.fhb.gov.hk/english/welcome/welcome.html.)

Sincerely,

Authors

 

Response to the Journal requirements 

Comment 1 1. Please ensure that your manuscript meets PLOS ONE's style requirements, including those for file naming. The PLOS ONE style templates can be found at https://journals.plos.org/plosone/s/file?id= wjVg/ PLOSOne_formatting_sample_ main_body.pdf and https://journals.plos. org/ plosone/s/file?id=ba62/PLOSOne_formatting_sample_title_authors_ affiliations.pdf

Response 1 We have changed the format of the manuscript and ensured that the manuscript meets the requirement of PLOS ONE. 

Comment 2 2. We note that the grant information you provided in the ‘Funding Information’ and ‘Financial Disclosure’ sections do not match. 

Response 2 Thank you for pointing out the inconsistent information. We have corrected the Funding information in the manuscript (Page 23, line 13). 

Comment 3 3. We note that you have indicated that data from this study are available upon request. PLOS only allows data to be available upon request if there are legal or ethical restrictions on sharing data publicly. For more information on unacceptable data access restrictions, please see http://journals.plos.org/plosone/s/data-availability#loc-unacceptable-data-access-restrictions. 

Response 3 Thanks for the comment. We have contacted with the Funder and then update our Data Availability statement as below:

All data are belong to the funder (Health and Medical Research Fund, Food and Health Bureau, Hong Kong Special Administrative Region). The data can be accessed only with the permission by the Bureau (https://rfs2.fhb.gov.hk/english/welcome/welcome.html.)

Comment 4 4. Please review your reference list to ensure that it is complete and correct. If you have cited papers that have been retracted, please include the rationale for doing so in the manuscript text, or remove these references and replace them with relevant current references. Any changes to the reference list should be mentioned in the rebuttal letter that accompanies your revised manuscript. If you need to cite a retracted article, indicate the article’s retracted status in the References list and also include a citation and full reference for the retraction notice.

Response 4 We have checked the reference list and ensure that it is complete and correct. 

Response to comments from Reviewer

Comment 1 1- Tables 1 to 3: For each baseline characteristic, provide only one p-value for comparing all three groups. If the p-value is significant, perform step-down tests to determine pairwise differences.

Response 1 Following the reviewer’s suggestion, the one-way ANOVA analysis was used to compare the differences of baseline variables among the three groups, and then a Tukey's honestly significant difference (HSD) post hoc test was performed to specific groups difference for the variables showing significant difference in the one-way ANOVA analysis. Please find the results in the table 1 to 3. The description of the statistical analysis method was revised accordingly as below (Page 9, line 8):

“For the variables showing significant difference in ANOVA, the Tukey's honestly significant difference post-hoc test the Games -Howell post-hoc test and were performed when the assumption of equal variances was met and not met, respectively.”

In addition, we will keep the standardized differences in table 1 and table 2 as they may allow the readers to well understand that demographic and smoking profiles among the three groups were well balanced after the PSM. 

Comment 2 2-Page 9: Explain the rationale for using a GEE model when adjusting for baseline characteristics rather than applying multivariate logistic regression models.

Response 2 We are apologized for the mistake of the description. Actually we used a Generalized Logistic Mixed Model (GLMM) to calculate the odds ratios as the data in this study were collected in multiple research center and the GLMM allowed us to adjust the random effect of the hospitals in the model. We have corrected the description in the manuscript as below (Page 9, line 20) and the notes under the table 4 accordingly: 

“A Generalized Logistic Mixed Model (GLMM) was then used to calculate the adjusted odds ratios (aORs) for primary and secondary outcomes after adjusting for characteristics at baseline and the random effect of hospitals using the matched sample.”

Comment 3 3- Page 10, Line 1: The figure number had been omitted.

Response 3 Thank you for pointing out the mistake, we have revised it as “Figure 1” in Page 10, line 1. 

Comment 4 4- Provide a post-hoc power calculation.

Response 4 Thank you for your suggestion, the results of power calculations have been added in the manuscript, please find it in table 5 and page 18, lines 1-4 as below:

“The observed power (1-β) of quitting immediately and quitting progressively on the biochemically validated quit rate, the self-reported quit rate, and self-reported reduction of cigarette consumption were then calculated using G*power.”

Comment 5 5- Explain the graphical results in figure 2.

Response 5 The explanation of the results in figure 2 has been added in the results section as below (Page 19, line 1-5):

“The scatterplot and ﬁtted line analysis showed that the values of both the absolute and percent cigarette reduction at the 1-, 3-, and 6-month follow-ups were associated with the biochemically validated abstinence as assessed at the 12-month follow-up (Fig 2). The R2 showed that the percent cigarette reduction (a-2, b-2, c-2) could better predict the 12-month abstinence than the absolute cigarette reduction (a-1, b-1, c-1).”

---

## [Decision Letter · Decision Letter 1]

4 Jul 2022

PONE-D-21-03778R1Effect of Quitting Immediately vs Progressively on Smoking Cessation for Smokers at Emergency Department in Hong Kong: A Posteriori Analysis of a Randomized Controlled TrialPLOS ONE

Dear Dr. Li,

Thank you for submitting your manuscript to PLOS ONE. After careful consideration, we feel that it has merit but does not fully meet PLOS ONE’s publication criteria as it currently stands. Therefore, we invite you to submit a revised version of the manuscript that addresses the points raised during the review process.

We look forward to receiving your revised manuscript.

Kind regards,

Hao Xue

Academic Editor

PLOS ONE

Journal Requirements:

Reviewers' comments:

Reviewer's Responses to Questions

**Comments to the Author**

1. If the authors have adequately addressed your comments raised in a previous round of review and you feel that this manuscript is now acceptable for publication, you may indicate that here to bypass the “Comments to the Author” section, enter your conflict of interest statement in the “Confidential to Editor” section, and submit your "Accept" recommendation.

Reviewer #1: (No Response)

2. Is the manuscript technically sound, and do the data support the conclusions?

Reviewer #1: Yes

3. Has the statistical analysis been performed appropriately and rigorously? 

Reviewer #1: Yes

4. Have the authors made all data underlying the findings in their manuscript fully available?

Reviewer #1: Yes

5. Is the manuscript presented in an intelligible fashion and written in standard English?

Reviewer #1: Yes

6. Review Comments to the Author

Reviewer #1: Minor Revision:

Include the full details of the power calculation. The power calculation should include: sample size, alpha level (indicating one or two-sided), minimal detectable difference (plus the standard deviation when appropriate) and statistical testing method.

7. PLOS authors have the option to publish the peer review history of their article (what does this mean?). If published, this will include your full peer review and any attached files.

Reviewer #1: No

---

## [Author Response · Author response to Decision Letter 1]

5 Jul 2022

Dear Editor,

Re: "Effect of Quitting Immediately vs Progressively on Smoking Cessation for Smokers at Emergency Department in Hong Kong: A Posteriori Analysis of a Randomized Controlled Trial" (PONE-D-21-03778R1)

We sincerely thank the editor and reviewer’s very useful and constructive comments. We have made changes accordingly. Please also refer to the following point-by-point responses to the comments from the reviewer and editor. 

Please kindly let me know in case any responses are not clear or the information is not adequate to clarify concerns. Thanks for giving us an opportunity to revise the manuscript. 

Please also note that all data are belong to the funder (Health and Medical Research Fund, Food and Health Bureau, Hong Kong Special Administrative Region). The data can be accessed only with the permission by the Bureau (https://rfs2.fhb.gov.hk/english/welcome/welcome.html.)

Sincerely,

Authors

 

Response to comments from Reviewer

Comment 1 Minor Revision required:

Include the full details of the power calculation. The power calculation should include: sample size, alpha level (indicating one or two-sided), minimal detectable difference (plus the standard deviation when appropriate) and statistical testing method.

Response 1 Thanks for the reviewer’s suggestion. We have added the following information related to sample size calculation in the text:

“The sample size was calculated according to a previous trial [3] of a smoking reduction plus nicotine replacement therapy intervention involving 1154 Chinese adult smokers unwilling to quit smoking (biochemically validated quit rate of 4.4% [10 of 226] in the control group and 8.0% [74 of 928] in the intervention group at 6months). To detect a two-sided significant difference between groups with a power of 80% and significance level of 5%, the required sample size was estimated to be 1088 participants (544 in each group). Given an expected attrition rate of approximately 30% at the 6-month follow-up, the target was at least 1554 individuals (777 in each group). Between July 4, 2015 and March 17, 2017, 1571 smokers who presented at 4 major emergency departments consented to participate in this randomized controlled trial and were randomized into an intervention group (n = 787) and a control group (n = 784).”

---

## [Decision Letter · Decision Letter 2]

15 Nov 2022

PONE-D-21-03778R2Effect of Quitting Immediately vs Progressively on Smoking Cessation for Smokers at Emergency Department in Hong Kong: A Posteriori Analysis of a Randomized Controlled TrialPLOS ONE

Dear Dr. LI,

Thank you for submitting your manuscript to PLOS ONE. After careful consideration, we feel that it has merit but does not fully meet PLOS ONE’s publication criteria as it currently stands. Therefore, we invite you to submit a revised version of the manuscript that addresses the points raised during the review process.

Your manuscript has been reassessed by two reviewers, whose reports can be found below. As you will see from the comments, there remain some minor additions to your sample size/power justification which should be addressed before your manuscript is suitable for publication.

We look forward to receiving your revised manuscript.

Kind regards,

Katrien Janin

Staff Editor

PLOS ONE

Journal Requirements:

Reviewers' comments:

Reviewer's Responses to Questions

**Comments to the Author**

1. If the authors have adequately addressed your comments raised in a previous round of review and you feel that this manuscript is now acceptable for publication, you may indicate that here to bypass the “Comments to the Author” section, enter your conflict of interest statement in the “Confidential to Editor” section, and submit your "Accept" recommendation.

Reviewer #1: (No Response)

Reviewer #2: All comments have been addressed

2. Is the manuscript technically sound, and do the data support the conclusions?

Reviewer #1: Yes

Reviewer #2: Yes

3. Has the statistical analysis been performed appropriately and rigorously? 

Reviewer #1: Yes

Reviewer #2: Yes

4. Have the authors made all data underlying the findings in their manuscript fully available?

Reviewer #1: Yes

Reviewer #2: Yes

5. Is the manuscript presented in an intelligible fashion and written in standard English?

Reviewer #1: Yes

Reviewer #2: Yes

6. Review Comments to the Author

Reviewer #1: Minor revision:

Sample Size/Power justification: State the statistical testing method which attains 80% power. Perhaps the method is a chi-square test for comparing proportions.

Reviewer #2: This is a well written articles and novel approach to smoking cessation, building on the authors previous works. Findings are clearly described and presented.

7. PLOS authors have the option to publish the peer review history of their article (what does this mean?). If published, this will include your full peer review and any attached files.

Reviewer #1: No

Reviewer #2: No

---

## [Author Response · Author response to Decision Letter 2]

18 Nov 2022

Thanks for your suggestion. We have added the following information to justify the sample size calculation in the text:

“The sample size was calculated according to a previous trial [3] of a smoking reduction plus nicotine replacement therapy intervention involving 1154 Chinese adult smokers unwilling to quit smoking (biochemically validated quit rate of 4.4% [10 of 226] in the control group and 8.0% [74 of 928] in the intervention group at 6months). To detect a two-sided significant difference between groups by a chi-square test for comparing proportions with a power of 80% and significance level of 5%, the required sample size was estimated to be 1088 participants (544 in each group). Given an expected attrition rate of approximately 30% at the 6-month follow-up, the target was at least 1554 individuals (777 in each group). Between July 4, 2015 and March 17, 2017, 1571 smokers who presented at 4 major emergency departments consented to participate in this randomized controlled trial and were randomized into an intervention group (n = 787) and a control group (n = 784).”

---

## [Decision Letter · Decision Letter 3]

12 Jan 2023

Effect of Quitting Immediately vs Progressively on Smoking Cessation for Smokers at Emergency Department in Hong Kong: A Posteriori Analysis of a Randomized Controlled Trial

PONE-D-21-03778R3

Dear Dr. Li,

We’re pleased to inform you that your manuscript has been judged scientifically suitable for publication and will be formally accepted for publication once it meets all outstanding technical requirements.

Kind regards,

Yann Benetreau

Staff Editor

PLOS ONE

Additional Editor Comments (optional):

Reviewers' comments:

Reviewer's Responses to Questions

**Comments to the Author**

1. If the authors have adequately addressed your comments raised in a previous round of review and you feel that this manuscript is now acceptable for publication, you may indicate that here to bypass the “Comments to the Author” section, enter your conflict of interest statement in the “Confidential to Editor” section, and submit your "Accept" recommendation.

Reviewer #1: All comments have been addressed

Reviewer #2: All comments have been addressed

2. Is the manuscript technically sound, and do the data support the conclusions?

Reviewer #1: (No Response)

Reviewer #2: (No Response)

3. Has the statistical analysis been performed appropriately and rigorously? 

Reviewer #1: (No Response)

Reviewer #2: (No Response)

4. Have the authors made all data underlying the findings in their manuscript fully available?

Reviewer #1: (No Response)

Reviewer #2: (No Response)

5. Is the manuscript presented in an intelligible fashion and written in standard English?

Reviewer #1: (No Response)

Reviewer #2: (No Response)

6. Review Comments to the Author

Reviewer #1: (No Response)

Reviewer #2: (No Response)

7. PLOS authors have the option to publish the peer review history of their article (what does this mean?). If published, this will include your full peer review and any attached files.

Reviewer #1: No

Reviewer #2: No

---

## [Editor Report · Acceptance letter]

17 Jan 2023

PONE-D-21-03778R3 

Effect of Quitting Immediately vs Progressively on Smoking Cessation for Smokers at Emergency Department in Hong Kong: A Posteriori Analysis of a Randomized Controlled Trial 

Dear Dr. Li:

I'm pleased to inform you that your manuscript has been deemed suitable for publication in PLOS ONE. Congratulations! Your manuscript is now with our production department. 

Kind regards, 

on behalf of

Katrien Janin 

Staff Editor

PLOS ONE